# Translated Benchmarks Can Be Misleading: the Case of Estonian Question Answering

**Hele-Andra Kuulmets    Mark Fishel**

Institute of Computer Science

University of Tartu

{hele-andra.kuulmets, mark.fisel}@ut.ee

## Abstract

Translated test datasets are a popular and cheaper alternative to native test datasets. However, one of the properties of translated data is the existence of cultural knowledge unfamiliar to the target language speakers. This can make translated test datasets differ significantly from native target datasets. As a result, we might inaccurately estimate the performance of the models in the target language. In this paper, we use both native and translated Estonian QA datasets to study this topic more closely. We discover that relying on the translated test dataset results in overestimation of the model's performance on native Estonian data.

## 1 Introduction

Translating test datasets to the target language has become a popular alternative to creating datasets from scratch in the target language (Yang et al., 2019, Ponti et al., 2020, Conneau et al., 2018). The main reason for this is that translating data, either manually or automatically, and reannotating it is easier than hiring data annotators to annotate the data. In addition, to ensure the quality of the newly created dataset, the authors often go through an exhaustive process of verifying the data quality, making creating new datasets even more expensive. On the other hand, existing datasets are already established in the NLP community. Another benefit of translated datasets is that they make evaluating cross-lingual transfer learning easier, as the identical datasets make the results directly comparable across languages.

However, in case only a translated test dataset exists for a specific task in a specific language, it is also likely true that there is probably no task-specific native training data available in that language. If there was native training data available, then a small subset of it could have been used to create a test dataset. Creating only a training dataset with no target test dataset available would also provide no benefit to the creators.

The existence of (translated) test dataset in some specific language, together with the non-existence of training data in the same language, has created an interesting situation where translated datasets have been mostly employed to advance cross-lingual transfer learning or related methods (e.g. TRANSLATE-TEST).[1] However, this contradicts the idea of these methods, which is to generalize to languages where training data for the task is unavailable. With translated test datasets, the training data *is* usually available[2]; it is just in another (source) language. In fact, it is most likely used to train the model, which will be evaluated with the translated test dataset. Because of this, there is a danger that evaluation results become artificially inflated and overestimate the model's performance on native data.

This paper aims to study the concerns of using translated test datasets more closely. We use English as a source language and Estonian as a target language and evaluate models trained on the source language with native and translated target datasets to see how the results on translated dataset compare to the results on the native dataset. We opt for TRANSLATE-TEST setup because it can be generalized more easily to different tasks as only a model trained in English is needed. In addition, it is competitive or even better at solving Estonian language understanding tasks than cross-lingual transfer methods (see Table 1).

---

[1] Some translated datasets, e.g XQuAD (Artetxe et al., 2020b) are specifically created to advance cross-lingual transfer research. Although the purpose of translating the dataset may differ, the outcome has the same issues that are addressed in this paper.

[2] Only test or validation split is usually translated.

| Dataset | Task | Metric | TRTE | TRTR | CL | Native | SOTA |
|---------|------|--------|------|------|-----|--------|------|
| EstQA (Käver, 2021) | extractive QA | F1 | 73.0 | **79.9** | 73.4 | 49.20 | 82.4 |
| News Stories (Härm and Alumäe, 2022) | abstractive summarization | ROUGE-1 | **17.22** | 17.0 | - | 16.22 | TRTE |
| XCOPA ET (Ponti et al., 2020) | commonsense reasoning | accuracy | **81.0** | 57.4[†] | 79.0[‡] | - | TRTE/81.0[‡] |

Table 1: Comparison of different methods on solving Estonian language understanding tasks. **TRTE**: TRANSLATE-TEST; **TRTR**: TRANSLATE-TRAIN; **CL**: cross-lingual transfer learning; **Native**: only native data was used for training; **SOTA**: reported state-of-the-art in literature (arbitrary method). Results are reported by the authors of the datasets if not specified otherwise: [†] Ruder et al. (2021); [‡] Muennighoff et al. (2022).

## 2 Related Work

TRANSLATE-TEST and TRANSLATE-TRAIN are commonly used machine translation baselines for cross-lingual transfer learning studies. (Conneau et al., 2018, Ponti et al., 2020, Lin et al., 2022, Hu et al., 2020, Liu et al., 2019). Somewhat surprisingly, TRANSLATE-TEST has shown to be a superior method for many languages in a cross-lingual setting where target language training data is not available (Ponti et al., 2020, Lin et al., 2022). Meanwhile, TRANSLATE-TRAIN has also been shown to outperform cross-lingual transfer learning methods and can compete with TRANSLATE-TEST (Ruder et al., 2021).

The success of machine translation-based methods has motivated researchers to improve these methods even more. Yu et al. (2022) shows that TRANSLATE-TRAIN can be improved by learning a mapping from originals to translationese that is applied during test time to the originals of the target language. Dutta Chowdhury et al. (2022) employs a bias-removal technique to remove translationese signals from the classifier. Oh et al. (2022) proposes TRANSLATE-ALL - a method that uses both techniques simultaneously. Their model is trained both on data in the source language and source data translated to the target language. During inference, the two predictions, one on the target dataset and another on the target dataset translated to the source language, are ensembled. Isbister et al. (2021) shows that even if a training dataset is available in the target language, it might still be beneficial to translate both training and test datasets to English to employ pre-trained English language models instead of native language models. Artetxe et al. (2020a) draws attention to the fact that even human-translated datasets can contain artifacts that can hurt the performance of the model when compared to the native English datasets. He shows that the performance drop is indeed caused by the fact that training is done on the original data while testing is done on translated data.

## 3 Methodology

Our goal is to compare evaluation results obtained with native and translated Estonian question-answering datasets in a TRANSLATE-TEST setting where the data is machine translated to English and fed to a model also trained on English. We hypothesize that translated test dataset will overestimate results on the native test dataset.

### 3.1 Models

**XLM-RoBERTa (Conneau et al., 2020)** A multilingual encoder trained on 100 languages (including Estonian) with masked language modeling objective. We fine-tune the base model XLM-ROBERTA-BASE.[3]

### 3.2 Datasets

**SQuAD (Rajpurkar et al., 2016)** An English extractive question-answering dataset consisting of more than 100 000 crowdsourced question-answer-paragraph triplets. The paragraphs are from English Wikipedia.

**XQuAD (Artetxe et al., 2020b)** A cross-lingual extractive question-answering benchmark that consists of 1190 triplets from SQuAD's validation set translated to 10 languages (not including Estonian) by professional translators. Each question has exactly one correct answer.

---

[3]https://huggingface.co/xlm-roberta-base

**EstQA (Käver, 2021)** An Estonian extractive question-answering dataset consisting of 776 train triplets and 603 test triplets where each question in the test dataset has possibly more than one correct answer. The paragraphs are from Estonian Wikipedia. It was specifically created to be an Estonian equivalent for English SQuAD.

### 3.3 XQuAD$_{et}$

We also need a translated Estonian question-answering dataset to see whether our hypothesis is true. This dataset should ideally be created using the same methodology as was used for the native dataset EstQA to avoid a situation where the difference in results could be attributed to different methodologies. Since EstQA was created by following the methodology used for SQuAD and XQuAD is a subset of it, we decided to translate the English subset of XQuAD to Estonian. The translation was done with Google Cloud API. The annotation spans were first automatically aligned with SimAlign (Jalili Sabet et al., 2020). After that, the alignments were verified manually, and corrections were made if necessary. We denote this dataset as **XQuAD$_{et}$**. Similarly to XQuAD, it consists of 1190 triplets.

### 3.4 Training and Inference

We train our QA model by fine-tuning XLM-ROBERTA-BASE SQuAD dataset. Ideally, we would have used existing QA models as this is one of the main benefits of the TRANSLATE-TEST approach. However, since XQuAD is a subset of the validation set of SQUAD, then this would have given an unfair advantage to XQuAD in our experiments.

During inference, the input (in Estonian) is machine translated to English using Google Cloud API and fed to a model trained on SQUAD. The predicted span (in English) is then automatically aligned with the input in Estonian using SimAlign to project the prediction back to Estonian.

### 3.5 Evaluation

Following Rajpurkar et al. (2016) we evaluate our models with exact match (EM) and f1 score (F1). Exact match is a metric that measures the percentage of predictions that match any of the gold labels exactly while F1 measures the average overlap between the predicted and gold answer. We use the

| Train data | Test data | EM | F1 |
|---|---|---|---|
| SQuAD | XQuAD$_{et}$ | **58.74** | **72.26** |
| | EstQA | 57.04 | 70.35 |

Table 2: TRANSLATE-TEST results on Estonian QA datasets.

| Train data | Test data | EM | F1 |
|---|---|---|---|
| EstQA | EstQA$_{en}$ | **26.37** | 41.99 |
| | XQuAD | 24.21 | **43.64** |

Table 3: TRANSLATE-TEST results on English QA datasets.

official scoring script of SQuAD.[4]

## 4 Results

Table 2 summarizes the main results of our experiments. The results support our hypothesis that using translated test datasets together with TRANSLATE-TEST can lead to overestimating the performance on the native target data. Note that in order to obtain the predictions for **XQuAD$_{et}$** the data was machine translated twice (first to Estonian and then during the inference back to English) but is still more easily solvable, despite the potentially stacking translation errors that can diminish the meaning of the texts.

### 4.1 Symmetry Test

We conducted an additional experiment to see whether our hypothesis is also true in the opposite direction, i.e., the model is trained on Estonian data and English test data is translated to Estonian during the inference. For that purpose, EstQA was translated to English using the same pipeline as for **XQuAD$_{et}$**. However, the results shown in Table 3 do not provide clear evidence that our hypothesis is also true in the opposite direction. Additionally, it can be seen that the results on both datasets are very low, which is expected since the EstQA training dataset contains only 776 training samples.

### 4.2 Quality of Automatic Annotations

The pipeline of solving QA task with TRANSLATE-TEST consists of multiple components, all of which work with some error rate. We can not assess the quality of machine-translated datasets because we do not have gold translations. However, both **XQuAD$_{et}$** and

---

[4]More precisely, we use `evaluate` library that wraps the original scripts: https://github.com/huggingface/evaluate.

| Dataset | EM | F1 |
|---------|------|------|
| EstQA$_{en}$ | 64.30 | 83.67 |
| XQuAD$_{et}$ | 83.61 | 91.40 |

Table 4: Annotation quality of automatic annotations.

**EstQA$_{en}$** contain human-verified annotations which we can compare against automatically obtained annotations. Table 4 shows the quality of automatic alignments on translated test datasets as measured with EM and F1 against manually corrected annotations. As the table shows, automatic alignments were much better for translated XQuAD, especially when comparing EM scores with nearly 20% difference.

The aligner algorithm in all our experiments was IterMax from the SimAlign package with a distortion of 0.5, as suggested by the authors. We used embeddings from BERT-BASE-MULTILINGUAL-CASED[5] (Devlin et al., 2019) as this yielded the best results in our experiments when compared to other contextual embeddings (see Appendix A for more details).

## 5 Discussion

### 5.1 Machine vs Human Translated Datasets

One may argue that in order to show that translated datasets are inferior to native datasets, human-translated data should be used instead of machine-translated data because usually translated datasets are created with the help of professional translators. However, we believe that it is not necessary. Firstly, it has been shown that regardless of the method, translated data contains translationese which makes it different from native data (Volansky et al., 2013, Bizzoni et al., 2020). Secondly, the cultural knowledge incorporated into the translated datasets will make them differ from native data despite the translation method. Finally, our goal was to investigate whether the model's performance would be overestimated with translated test datasets. Intuitively, this is more difficult to show with machine-translated data because of potential translation errors. Therefore, if the hypothesis is true with machine-translated data, it is fair to assume that it will also be true for human-translated data.

[5]https://huggingface.co/bert-base-multilingual-cased

### 5.2 Cause of Mismatch

The problem we are addressing in this paper is caused by the fact that data from the same distribution is often used to train and evaluate models in a TRANSLATE-TEST setting where cultural differences of languages should naturally be taken into account. However, one may say that this argumentation leads to the same conclusions about monolingual research because it also uses different splits of the same dataset for training and testing. Although domain shift is a problem in monolingual research, it differs from the scenario addressed in this paper. Domain mismatch happens because the model learns to detect unwanted biases in the training dataset that are irrelevant to solving the task in general (McCoy et al., 2019, Jia and Liang, 2017). The mismatch in our scenario happens because different cultural knowledge is naturally intertwined into each of the languages by the speakers, which the model trained only on one language can not know about.

### 5.3 Asymmetry

Our experiments showed that overestimating happens when native Estonian data is translated to English but not when native English data is translated to Estonian during test-time data augmentation, i.e., not always are translated datasets easier to solve for the model. However, the results might also be affected by the properties of the underlying language model or train dataset size. For a more fair comparison of translation directions, the train datasets should be around the same size. Currently, the difference in sizes is more than 100 times.

### 5.4 Limitations

The main limitation of the paper is its relatively small scale which can be overcome by including more languages, more datasets, or a cross-lingual transfer scenario. Alternatively, one can translate test datasets from languages other than English to Estonian (or any other target language) and compare the performance in TRANSLATE-TEST (et → en) setup.

## 6 Conclusion

We compared the performance of an English extractive QA model on native and translated Estonian test datasets in TRANSLATE-TEST setting to see how results on the translated dataset compare

to the results on the native dataset. Our experiments showed that results on the translated dataset overestimate the results on the native dataset.

## Acknowledgements

This article has been financed/supported by European Social Fund via "ICT programme" measure.

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

# A Performance of SimAlign with different embeddings

Since the authors of SimAlign did not evaluate their choice of embedding on Estonian, we did our own evaluation with three different embeddings. Figure 1 and Figure 2 show how the choice of embedding affects the quality of alignments.

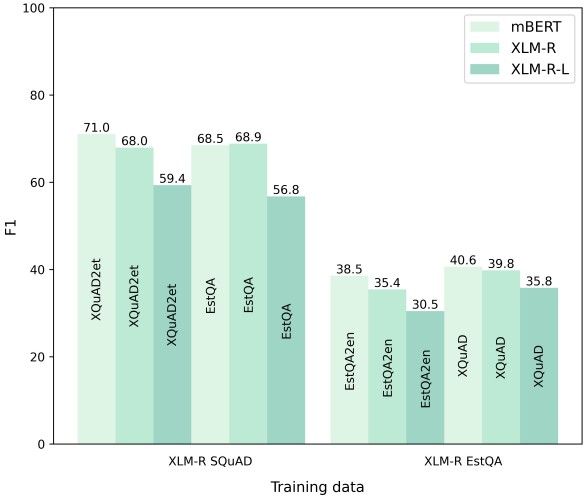

Figure 1: F1 of automatically aligned answers with different embeddings.

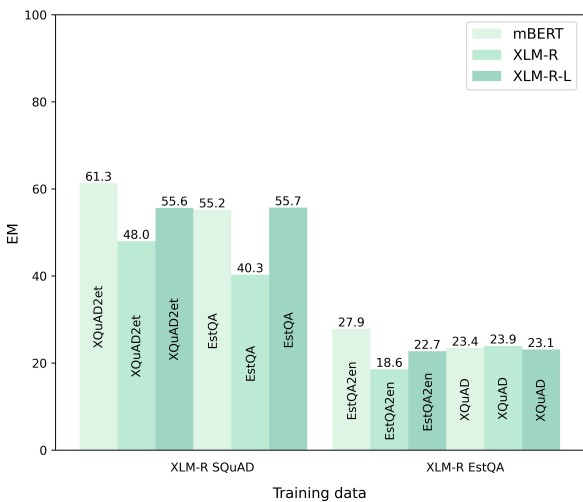

Figure 2: EM of automatically aligned answers with different embeddings.

The scores are obtained by comparing predictions projected back to the target language with gold annotations. As the authors of SimAlign, we found that embeddings from mBERT produce the best alignments. Note that the scores obtained with mBERT are not the same as shown in Table

2. This is because the algorithm that projected predicted spans back to the target language was slightly changed before obtaining the final results.

# B Hyperparameters

For both English and Estonian QA models, XLM-R was fine-tuned with learning rate $2e^{-5}$ (linear decay) and batch size 16 for 20 epochs with early stopping after ten consecutive evaluation steps with no improvement in validation loss. The model was evaluated after every 100 steps. Weight decay was 0.01, warmup ratio 0.