# OpenReview forum: "Translated Benchmarks Can Be Misleading: the Case of Estonian Question Answering"
_NoDaLiDa/2023/Conference — NoDaLiDa 2023_

### Official Review · Reviewer_bLDN · 2023-03-08
**Comparison of translated to original benchmarks for QA**

**Rating:** 7
**Confidence:** 3

**Review:**

This paper deals with the area of (machine) translated QA datasets. Given the recent popularity of machine translated datasets, it is naturally important to ask what the limitations of such approaches are. The paper focuses on the translate-test approach. While interesting and commonly studied, it is the less practical of the two approaches considered, since test-time translation is quite unrealistic in a real-world deployment.

This short paper is a little difficult to follow, since the narrative is quite compressed and needs very careful reading and re-rereading to make sure I understand correctly what was done, which data was translated from which language and to which language at which point, etc. It was not an easy read. Perhaps some systematic and consistent way of marking each mention of a dataset with the source and target language, and repeating the source and target languages also systematically in the tables would help the reader confidently follow the paper.

The main results in Table 2 are not discussed in a particularly detailed manner. While the direction of the difference seems to support the original hypothesis for the most part, the difference, numerically, is not particularly large. I think this would deserve a more thorough discussion. Has the experiment, after all, fully confirmed the hypothesis?

All in all, despite the somewhat difficult read, I think this is a good and relevant contribution to NoDaLiDa



**Paper Type:**

Short paper

---

### Official Review · Reviewer_TU7q · 2023-03-09
**The paper addresses the differences in using native vs. translated test sets in extractive Question Answering**

**Rating:** 7
**Confidence:** 4

**Review:**

The paper discusses the effects of using a native test set or a translated test set when generating benchmarks for extractive Question Answering.
The subject can be positioned in the global context in NLP of how low resource languages can benefit from large data sets, models and methods developed for and with English but at the same time the need not to over generalize from those resources to every language.
The paper is well structured and has a clear scope. For people not all to familiar with the subject it would help to add short definitions to some terms and abbreviations, like EM, and e.g. explain the meaning of the F1 score in the context of extractive QA.
As for the results, where the authors claim that different cultural knowledge is the cause of mismatch in the answer extraction, it would be good to see evidence/examples of that claim from the evaluation results.

**Paper Type:**

Short paper

---

### Official Review · Reviewer_jCr7 · 2023-03-10
**Translate QA benchmarks for Estonian Evaluation**

**Rating:** 7
**Confidence:** 3

**Review:**

In the paper "Translated Benchmarks Can Be Misleading: the Case of Estonian Question Answering" the authors look at the performance difference between a translated test set and a native Estonian test set for QA.
For this they take the multilingual XLM-R model, train it on English data  SQuAD and evaluate it on a translated XQuAD and native EstQA.

At test time the Estonian data is translated to English, the model predicts the span for the answer, which is then aligned to the original Estonian data to produce the final prediction.
This approach is interesting for languages that do not have strong monolingual base language models or sufficient representation in a multilingual model.
In their evaluation, the authors show that the model performs somewhat better on the English-to-Estonian translated data than on the native Estonian data.
To test whether this behaviour, translated test data results overestimate native data results, is also true for Estonian-English, the authors train the minimally-Estonian XLM-R on the much smaller EstQA data to the evaluate on native English and translated-to-English data.
These results do not mirror the previous (for F1), and are much worse as also noted by the authors due to the aforementioned problems and the lower quality of automatic annotation/alignment compared to the main experiment.

## Pros
- the goals are clearly presented
- the writing is mostly easy to follow and understandable for non-QA experts
- the pipeline is well described and would help other researchers if shared

## Cons
- the title is a bit broad given one experimental result with a 2% f-score difference
- fine-tuning LMs such as XLM-R can lead to varying performance given different hyperparameters but also different random seeds
- I am not familiar with performance on XQuAD in general, but averaged scores over multiple runs or statistical significance tests would make the performance difference more impactful
- the authors note themselves that their symmetry test is to be taken with a big spoon of salt,  making it somewhat unnecessary


## Other
- Will the pipeline be published?
- Will the translated and aligned XQuAD be published?
- EM most likely refers to exact match and F1 to f-score; please introduce your measures at least in passing

**Paper Type:**

Short paper

---

### Decision · Program_Chairs · 2023-03-17

Accept